# Spatiotemporal dynamics of moiré excitons in van der Waals heterostructures

Giuseppe Meneghini ⬡ ✉, Samuel Brem & Ermin Malic ⬡

Heterostructures of transition metal dichalcogenides (TMDs) offer unique opportunities in optoelectronics due to their strong light-matter interaction and the formation of dipolar interlayer excitons. Introducing a twist angle or lattice mismatch between layers creates a periodic moiré potential that significantly reshapes the energy landscape and introduces a high-dimensional complexity absent in aligned bilayers. Recent experimental advances have enabled direct observation and control of interlayer excitons in such moiré-patterned systems, yet a microscopic theoretical framework capturing both their thermalization and spatiotemporal dynamics remains lacking. Here, we address this challenge by developing a predictive, material-specific many-body model that tracks exciton dynamics across time, space, and momentum, fully accounting for the moiré potential and the complex non-parabolic exciton band structure. Surprisingly, we reveal that flat bands, which typically trap excitons, can significantly enhance exciton propagation. This counterintuitive behavior emerges from the interplay between the flat-band structure giving rise to a bottleneck effect for exciton relaxation and thermal occupation dynamics creating hot excitons. Our work not only reveals the microscopic mechanisms behind the enhanced propagation but also enables the control of exciton transport via twist-angle engineering. These insights lay the foundation for next-generation moiré-based optoelectronic and quantum technologies.

In the past decade, heterostructures built from transition metal dichalcogenides (TMD) have gained a lot of attention as a flexible and tunable platform for investigating a variety of many-particle phenomena[1–8]. Among the most interesting observations are the emergence of strongly correlated phases, ranging from Mott insulating behavior to Wigner crystals, alongside unconventional exciton transport[9–15]. In addition, TMD heterostructures typically exhibit a type-II band alignment, leading to the formation of interlayer excitons, characterized by a permanent out-of-plane dipole moment[16–19]. Together, these effects reveal the remarkable degree of control offered by van der Waals interfaces and open new directions for studying and manipulating quantum states in low-dimensional systems.

A central mechanism underlying many of these phenomena is the moiré potential, which emerges due to a lattice mismatch or a finite twist angle between the two layers building a heterostructure[2,20–22]. This long-range periodic modulation drastically reshapes the energy landscape, leading to the formation of exciton subbands in the mini-Brillouin zone (mBZ), resulting in a much more intricate band structure than in the untwisted TMD bilayer[23]. The additional periodicity has been shown to significantly alter optical selection rules, with new resonances appearing in absorption spectra[23–26]. Moreover, the strength of the moiré potential has a pronounced impact on exciton transport[27]: depending on the twist angle, diffusion can vary from complete suppression, associated with the emergence of flat exciton bands at small angles[28,29], to anomalous propagation regimes[30].

While various approaches have been proposed to describe exciton diffusion in moiré materials, most rely on simplified assumptions, typically treating energy relaxation and spatial diffusion as decoupled

Department of Physics, Philipps University of Marburg, Marburg, Germany. ✉e-mail: giuseppe.meneghini@physik.uni-marburg.de

processes[31,32]. These approximations, although insightful, fail in capturing the coupled dynamics that arise when excitons simultaneously exhibit a non-trivial band structure, efficient energy relaxation via phonon scattering, and spatial localization due to the moiré potential. A comprehensive theoretical framework capable of microscopically addressing the interplay between energy relaxation and real-space diffusion has remained unexplored. Bridging this gap is not only critical for advancing our fundamental understanding of exciton transport in moiré systems but also essential for enabling future optoelectronic applications that exploit moiré engineering to control the exciton flow. In this work, we take an important step in this direction by developing a microscopic model that captures both the momentum and spatial dynamics of moiré excitons. By solving the Boltzmann transport equation with full momentum and spatial resolution, we reveal a counterintuitive regime of exciton transport in moiré materials: despite the presence of flat bands that are expected to hinder motion, exciton diffusion is predicted to be significantly enhanced at low temperatures. This unexpected behavior arises from an interplay between the moiré band structure and phonon-mediated relaxation processes, highlighting a new mechanism of efficient energy transport in systems with strong periodic potentials.

## Results

### Microscopic model

We study the spatiotemporal moiré exciton dynamics in a twisted TMD heterostructure in the low excitation regime, where the exciton density remains low, so that exciton-exciton interactions can be neglected. Our approach is based on an equation-of-motion formalism[33–36]. A transformation to the Wigner representation results in a Boltzmann transport equation in the moiré exciton basis (Eq. (3) in the methods section). In this way, we can track the time evolution of the exciton distribution in momentum, energy, and space in the presence of a periodic moiré potential. In contrast to the case of free excitons characterized with a parabolic dispersion, where the thermal equilibrium is described by a Boltzmann distribution, the inclusion of the moiré potential drastically increases the complexity of the problem. The moiré-modified bands are not parabolic anymore, i.e., we cannot restrict the study to the solution of the radial component, but the full two-dimensional momentum-dependent band structure has to be taken into account. Furthermore, the number of relevant moiré exciton subbands within the thermal energy window increases substantially as the twist angle decreases. Moreover, the thermalization dynamics can exhibit pronounced relaxation bottleneck effects, leading to considerable deviations from the standard Boltzmann distribution. To be able to capture the full spatiotemporal moiré exciton dynamics, we solve the Boltzmann transport equation in both momentum and real space, employing a Monte Carlo algorithm[37–39] to manage the high dimensionality of the problem. This material-specific and microscopic framework allows us to analyze how the interplay of the moiré band structure and phonon-mediated relaxation channels governs the diffusion process. The developed theoretical framework is applicable to a larger class of moiré systems, including lattice-mismatched heterobilayers, by appropriately adjusting the mapping between the twist angle and the moiré potential strength. Key equations are presented in the Methods section, and further technical details are provided in the Supplementary Information.

### Moiré exciton diffusion

We apply our model to the exemplary twisted hBN-encapsulated $WSe_2$-$MoSe_2$ heterostructure. In this material, interlayer KK excitons are the lowest energy states, where the electron and hole reside in different layers[40,41]. This spatial separation gives rise to a permanent out-of-plane dipole moment and results in an extended exciton

lifetime[7,16–18,40–42]. Due to the weak interlayer tunneling near the K points, this state remains largely unaffected by layer hybridization[15,43]. As a consequence, the moiré potential can be effectively described by using a continuum model including hybridization in momentum space[23]. We initialize an exciton distribution with a Gaussian profile in real space, characterized by a standard deviation of 1 μm, and a uniform energy distribution of approximately 60 meV, focusing on an intermediate twist angle of 3°. At very small twist angles around 1°, exciton dispersion becomes completely flat and localizes excitons in the moiré potential minima[23]. Here, the group velocity of excitons becomes zero, inhibiting transport in the low-density regime, as observed experimentally[28,29] and predicted theoretically[32]. Therefore, in this work, we focus on an intermediate range of twist angles, where the moiré potential significantly modifies the excitonic band structure, but does not completely trap excitons.

By solving the Boltzmann transport equation (Eq. 3), we track the time- and space-dependent evolution of the exciton population. In particular, we study exciton mobility and quantify the impact of the moiré potential on spatial diffusion by extracting the diffusion coefficient $D$. The results are presented in Fig. 1, where we show time-resolved spatial profiles of the exciton distribution. Each profile is individually normalized to highlight the progressive broadening of the distribution over time. The insets illustrate the variance $\sigma^2$ of the exciton distribution as a function of time, with the slope determining the diffusion coefficient $D = \frac{1}{4}\partial_t\sigma_t^2$ [36]. We perform the simulations under identical initial conditions at two different temperatures. A quantitative comparison of the extracted diffusion coefficients reveals a distinct temperature dependence. At 70 K, we obtain $D = 1.4\,cm^2/s$, whereas at 10 K, the diffusion increases by almost a factor of 5 to $D = 6\,cm^2/s$.

### Diffusion coefficient analysis

To explore in detail the influence of temperature and energy relaxation on real-space exciton dynamics, we calculate the diffusion coefficient as a function of temperature, cf. Fig. 3(a). In particular, we compare our findings with the case of an ideal Boltzmann-distributed exciton population by using the analytical expression for the diffusion coefficient, obtained within the relaxation-time approximation (more details

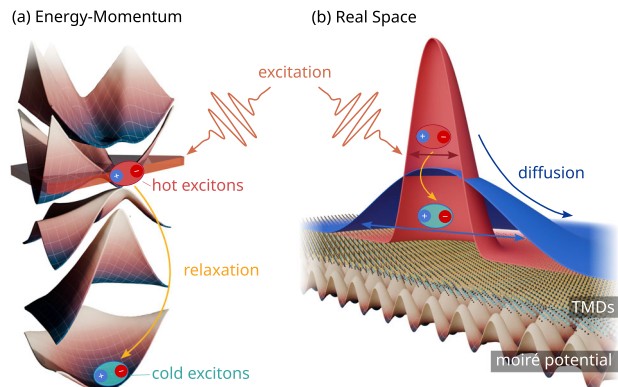

(a) Energy-Momentum    (b) Real Space

**Fig. 1 | Schematic of the system under study. a** A twisted TMD heterostructure is optically excited, creating an initial population of hot excitons (represented by the red contoured and shaded electron-hole pair bubble) with an energy of 60 meV (indicated by the orange plane pointed to by the arrow in the excitonic subband structure). **b** Spatial distribution of excitons with a 0.5 μm width (red Gaussian). The system undergoes an energy-momentum thermalization (orange arrows connecting red excitons (hot) to blue excitons (cold)), accompanied by time-dependent spatial diffusion, illustrated by the blue arrow denoting the broadening of the spatial exciton distribution (blue Gaussian). Both processes are strongly dependent on twist angle and temperature.

can be found in the Supplementary Information.)

$$D = \frac{1}{2} \sum_{\mathbf{k}\eta} |\mathbf{v}_{\mathbf{k}}^{\eta}|^2 \tau_{\mathbf{k}}^{\eta} N_{\mathbf{k}}^{\eta} \qquad (1)$$

with the moiré exciton distribution $N_{\mathbf{k}}^{\eta}$. The diffusion coefficient $D$ is governed by a competition between the squared group velocity $|\mathbf{v}_{\mathbf{k}}^{\eta}|^2$ and the scattering time $\tau_{\mathbf{k}}^{\eta}$, which has been obtained as the inverse of the total out-scattering rate from the state $\eta$ (see the Supplementary Information. for more details). At low temperatures, we predict significant deviations from the Boltzmann behavior, cf. the orange-shaded area in Fig. 3a. Although flat bands should suppress diffusion due to the vanishing group velocity, surprisingly, we observe an opposite trend: for temperatures below 70 K, we find a significant enhancement of exciton diffusion. The diffusion coefficient at 10 K reaches a value of $D \simeq 5.6$ cm²/s that is more than double the value of $D \simeq 2.7$ cm²/s expected for a Boltzmann distribution.

To gain further insight, we examine the stationary exciton distributions obtained at two representative temperatures of 10 K and 70 K. At lower temperatures, excitons remain trapped (bottleneck effect) in relatively flat regions of the dispersion landscape (cf. Fig. 3b, showing the exciton occupation superimposed on the band structure along the path $\gamma \rightarrow \kappa \rightarrow m$). Here, the mismatch between the interband energy gap and the energies of the dominant optical phonons prevents further relaxation to the ground state via phonon emission[44]. However, the excitonic band structure is not flat in all directions, and the thermal population can partially extend into more dispersive regions of the moiré Brillouin zone (cf. the schematic 3D plot of the excitonic band structure in Fig. 2, allowing excitons to access states with higher group velocities. This directional extension of the population explains the larger diffusion coefficient compared to the fully thermalized case, despite the apparent flatness seen along the plotted path. This is a counterintuitive result, as one might expect flat bands to trap excitons and hinder their propagation. The situation is considerably different at 70 K, where the ground state clearly has the largest occupation and the higher energy bands are only weekly occupied as expected from an equilibrium Boltzmann distribution (cf. Fig. 3c).

The flatness of exciton subbands plays a crucial role for the efficiency of possible phonon-driven scattering channels. Therefore, we investigate now the role of the twist angle that can be used as a tuning knob for the exciton band structure and thus also for the exciton propagation. Figure 4a displays the diffusion coefficient as a function of temperature for various twist angles. Two distinct regimes can be identified: one at higher temperatures (>50 K) and one at low temperatures (<50 K). In the high-temperature regime, the diffusion coefficient increases monotonically with the twist angle, approaching the asymptotic value expected for interlayer excitons with a parabolic band. In contrast, at lower temperatures, the presence of the moiré potential induces miniband gaps and flat-band regions of vanishing group velocity. The smaller the twist angle, the larger the energy gaps and the flatter the bands[23]. As a result, one would expect excitons to be at least partially trapped at lower twist angles, significantly reducing their diffusion. However, as already observed in Fig. 3a, exciton diffusion becomes considerably faster at the smallest considered twist angle of 3° and at low temperatures, cf. Fig. 4a. Moreover, for 3°, the diffusion coefficient decreases with increasing temperature, while higher twist angles exhibit a non-monotonic temperature dependence.

To understand these remarkable observations, we return to the analytical expression for the diffusion coefficient in Eq. (1). While the scattering time $\tau$ generally decreases monotonically with temperature, due to enhanced exciton-phonon scattering, the group velocity contribution increases, as higher-energy (steeper) regions of the bands become thermally accessible. Although both quantities contribute to the diffusion coefficient $D$ via a momentum and band-summed

integral, qualitative trends can be captured by introducing an effective group velocity squared $v_{eff}^2 = \sum_{\eta \mathbf{k}} v_{\eta \mathbf{k}}^2 N_{\mathbf{k}}^{\eta}$ and an effective scattering time $\tau_{eff} = \sum_{\eta \mathbf{k}} \tau_{\eta \mathbf{k}} N_{\mathbf{k}}^{\eta}$. These are shown in Fig. 4b, c as a function of temperature for different twist angles. While $\tau_{eff}$ increases monotonically with the twist angle, $v_{eff}^2$ becomes significantly enhanced at low temperatures in the case of 3°, due to the non-thermal exciton distribution caused by the bottleneck effect as shown in Fig. 3b. As thermal broadening activates scattering into lower-energy (flatter) regions of the band structure, the expected trend of increasing $v_{eff}^2$ with twist angle is recovered at higher temperatures.

The non-monotonic temperature dependence of the diffusion coefficient observed for twist angles above 3° arises from a subtle competition between two processes: thermal occupation favors higher-velocity states (increasing $v_{eff}^2$ with temperature), while enhanced phonon scattering reduces $\tau_{eff}$. For twist angles beyond the bottleneck regime (e.g., >3°), this interplay leads to a maximum in the diffusion coefficient at the temperature where the decrease in $\tau_{eff}$ begins to dominate the gain in $v_{eff}^2$. Moreover, the position of this maximum slightly shifts with the twist angle, which we attribute to twist-induced changes in the effective exciton mass. In contrast, the monotonic decrease in the diffusion coefficient for 3° can be understood by considering the exciton band structure in Fig. 3b. The band gap lies within the thermally populated region between 40 K and 70 K. The absence of available states in this energy window prevents the group velocity from compensating for the temperature-induced decrease in scattering time, leading to a sharp drop in the diffusion coefficient.

## Discussion

We have developed a microscopic framework based on Monte Carlo solutions of the Boltzmann transport equation to model time-, momentum-, and space-resolved exciton dynamics in the presence of a moiré potential. This unified approach captures both momentum-space thermalization and real-space diffusion. We apply the model to the exemplary case of a twisted hBN-encapsulated WSe₂-MoSe₂ heterostructure. We focus on a twist angle range of 3°–6°, where the strength of the moiré potential is not strong enough to fully localize excitons, but it still considerably modifies the exciton band structure. We predict an unexpected increase of exciton propagation at lower twist angles and low temperatures—in spite of the emergence of flat bands, which typically describe immobile excitons. We trace this surprising behavior to the relaxation bottleneck that prevents excitons to fully dissipate their excess energy. This leads to an accumulation of high-energy excitons close to flat-band regions. Here, one would naively expect a negligible spatial propagation, as the group velocity approaches zero. However, the thermal population of excitons extending beyond the flat regions results in larger effective group velocities and explains the increased exciton diffusion. At higher temperatures, this effect vanishes, since excitons are not trapped anymore, restoring a thermal distribution. This regime is qualitatively distinct from the fully localized case at smaller twist angles around 1°, exhibiting flat bands and trapped excitons, as well as from the parabolic regime at high twist angles, where excitons are mobile and no relaxation bottleneck appears, allowing them to form a thermalized Boltzmann distribution. The developed methods have been applied to an exemplary TMD heterostructure, however, they are applicable to a larger class of materials, including those heterostructures, where the moiré potential arises from a larger lattice mismatch rather than from the twist angle. Overall, our study provides new microscopic insights into exciton transport in moiré superlattices and lays the foundation for the design of optoelectronic devices, in which exciton diffusivity and propagation length can be tuned by varying the twist angle and temperature. This enables a control of the exciton flow that is important for e.g., excitonic circuits, energy funneling, or diffusion-mediated light emission.

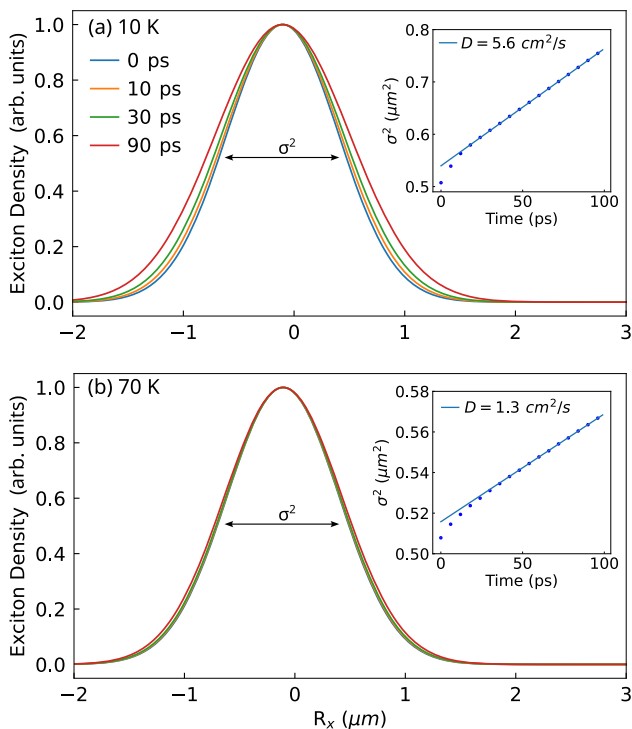

**Fig. 2 | Spatial distribution time evolution.** Time-dependent real-space cuts along the x direction ($R_x$) of the exciton distribution in an hBN-encapsulated WSe$_2$-MoSe$_2$ heterostructure for a twist angle of 3° and at a temperature of (**a**) 10 K and (**b**) 70 K. Insets show the variance $\sigma^2$ of the spatial distribution over time (blue dots). An initial non-linear increase (within the first tens of ps) corresponds to a higher diffusion coefficient reflecting the propagation behavior of transient hot excitons. At later times, we find a linear regime that is used to extract the stationary diffusion coefficient (blue solid lines).

## Methods

### Moiré exciton Hamiltonian

To describe a TMD heterostructure in the presence of a moiré potential (arising from twist angle or lattice mismatch) we start from the bilayer Hamiltonian[19,26], including intralayer and interlayer excitons and their interaction with phonons $H_X + H_{X-ph} = \sum_\mu \mathcal{E}_{\mathbf{Q}}^\mu X_{\mathbf{Q}}^{\mu\dagger} X_{\mathbf{Q}}^\mu + \sum_{\mu\nu j\mathbf{qQ}} D_{j\mathbf{qQ}}^{\mu\nu} X_{\mathbf{Q+q}}^{\nu\dagger} X_{\mathbf{Q}}^\mu b_{j\mathbf{q}} + h.c.$, with $\mathcal{E}_{\mathbf{Q}}^\mu$ denoting the free exciton energy, $X_{\mathbf{Q}}^{\mu\dagger}$ being the operator creating an exciton in the state $\mu$ (intra-/interlayer 1s state) with the center of mass momentum $\mathbf{Q}$, $D_{j\mathbf{qQ}}^{\mu\nu}$ describing the exciton-phonon matrix element, and $b_{j\mathbf{q}}$ corresponding to the phonon annihilation operator with the phonon mode $j$ and the momentum transfer $\mathbf{q}$. Note that we neglect the hybridization of intra- and interlayer exciton states, as the wavefunction overlap is known to be small at the K point[26,45–47]. Focusing on the excitonic ground state in the following, we neglect the excitonic index $\mu$.

We introduce the effect of the twist angle in terms of a continuum model for the moiré potential[2,22,23], $V_M = \sum_{\mathbf{Qg}} \mathcal{M}_{\mathbf{g}} X_{\mathbf{Q+g}}^\dagger X_{\mathbf{Q}}$ with $\mathbf{g} = s_1 \mathbf{G}_1^M + s_2 \mathbf{G}_2^M$, where $\mathbf{G}_{1/2}^M$ are reciprocal moiré lattice vectors and $s_{1/2}$ integers, with $\mathcal{M}_{\mathbf{g}}$ referring to the effective potential generated by the local displacement of the two twisted layers. This continuum model accurately captures the low-energy moiré exciton physics in the regime of small twist angles and weak interlayer tunneling, assuming that the moiré potential only slightly affects the intralayer exciton-phonon interaction, leading mainly to a remodulation of energy and momentum. More details can be found in the Supplementary Information. By applying a zone-folding procedure in the excitonic Hamiltonian we diagonalize the free exciton Hamiltonian with the moiré

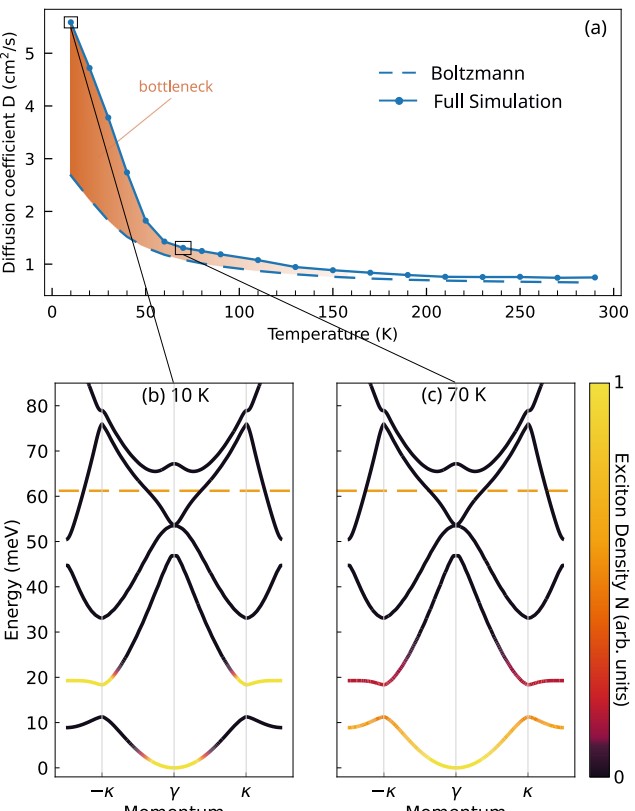

**Fig. 3 | Temperature-dependent diffusion coefficient. a** Temperature-dependent diffusion coefficient for hBN-encapsulated WSe$_2$-MoSe$_2$ heterostructure with a twist angle of 3°. Simulation results (solid-dotted line) are compared with the case assuming a Boltzmann distribution (dashed line). Strong deviations for temperatures lower than approximately 60 K arise due to a pronounced relaxation bottleneck. Exciton occupation at equilibrium (color-coded) overlaid on the exciton subband structure at a temperature of 10 K (**b**) and 70 K (**c**). At low temperatures, excitons are trapped in local dispersion minima far from the ground state. They exhibit a larger group velocity and thus lead to an effective increase in the diffusion coefficient compared to the Boltzmann case. At high temperatures, full relaxation is restored, and the simulation aligns well with the Boltzmann model. The orange dashed line represents the energy initialization condition.

potential term $H_X + V_M$, introducing new moiré exciton operators $Y_{\mathbf{Q}}^\eta = \sum_{\mathbf{g}} \omega_{\mathbf{g}}^\eta(\mathbf{Q}) X_{\mathbf{Q+g}}$[23], where now $\mathbf{Q}$ is the momentum in the mini Brillouin zone. This results in the full Hamiltonian for the system

$$\tilde{H} = \sum_\eta E_{\mathbf{Q}}^\eta Y_{\mathbf{Q}}^{\eta\dagger} Y_{\mathbf{Q}}^\eta + \sum_{\substack{\eta\xi j \\ \mathbf{QQ'g}}} \tilde{\mathcal{D}}_{\mathbf{QQ'g}}^{\eta\xi j} Y_{\mathbf{Q'}}^{\xi\dagger} Y_{\mathbf{Q}}^\eta b_{\mathbf{Q'-Q+g}}^j + h.c. \quad (2)$$

with $\tilde{\mathcal{D}}_{\mathbf{QQ'g}}^{\eta\xi j}$ as the exciton-phonon coupling tensor in the new basis containing the overlap of initial and final moiré states.

### Moiré exciton equation of motion

To be able to track the real and momentum space dynamics of moiré excitons, we derive the equation of motion for the off-diagonal terms in the moiré exciton density matrix formalism expressed in the Wigner representation, extending the approach introduced by Hess and Kuhn[33–36] $\tilde{f}_{\mathbf{k}}^\eta(\mathbf{r}) = \sum_{\mathbf{l} \in mBZ} e^{i\mathbf{l}\cdot\mathbf{r}} \langle Y_{\mathbf{k-l}}^{\eta\dagger} Y_{\mathbf{k}}^\eta \rangle$. We assume that the Wigner function has a slow envelope, i.e., the excitation area in real space is much larger than the moiré unit cell ($\mu$m of excitation spot against nm for the moiré unit cell). Thus, we can restrict to small momenta in the off-diagonal terms and obtain the Boltzmann transport equation for

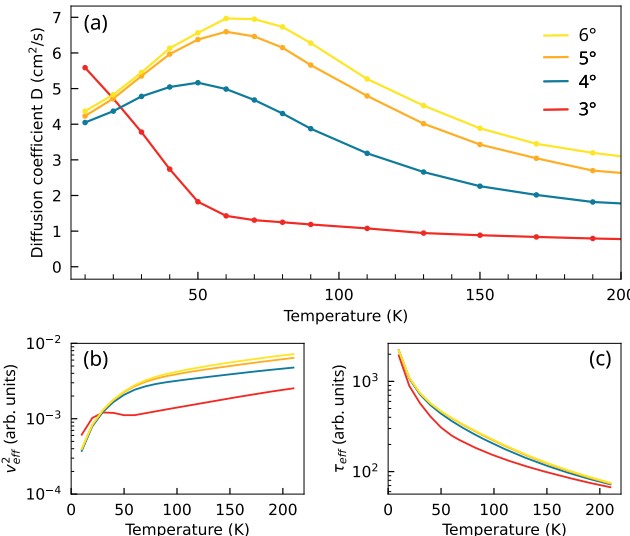

**Fig. 4 | Twist-angle-dependent exciton diffusion. a** Twist angle and temperature dependence of the diffusion coefficient. At temperatures higher than 60 K, the diffusion coefficient increases with the twist angle, driven by the concurrent rise in both the effective group velocity $v_{\text{eff}}^2$ and the effective scattering time $\tau_{\text{eff}}$ shown in (**b, c**) respectively. At low temperatures ($T < 50$ K) and for the smallest angle of 3°, the dynamics is dominated by the relaxation bottleneck giving rise to an occupation of hot excitons in higher exciton bands and as a consequence to an enhanced diffusion coefficient. In this regime, the squared effective group velocity $v_{\text{eff}}^2$ at 3° exhibits a non-monotonic dependence on temperature. Here, low-temperature velocity values exceed those of larger twist angles, highlighting the direct impact of the bottleneck on the effective excitonic group velocity.

moiré excitons, reading,

$$\dot{\tilde{f}}_{\mathbf{k}}^{\eta}(\mathbf{r}) = - \mathbf{v}_{\mathbf{k}}^{\eta} \nabla_{\mathbf{r}} \tilde{f}_{\mathbf{k}}^{\eta}(\mathbf{r}) + \sum_{\xi\mathbf{p}} \left[ W_{\mathbf{pk}}^{\xi\eta} \tilde{f}_{\mathbf{p}}^{\xi}(\mathbf{r}) - W_{\mathbf{kp}}^{\eta\xi} \tilde{f}_{\mathbf{k}}^{\eta}(\mathbf{r}) \right] \tag{3}$$

where $\mathbf{v}_{\mathbf{k}}^{\eta} = 1/\hbar \nabla_{\mathbf{k}} E_{\mathbf{k}}^{\eta}$ is the group velocity obtained from the moiré exciton dispersion, and $W_{\mathbf{kp}}^{\eta\xi}$ is the scattering tensor encoding the moiré exciton-phonon scattering elements. The full derivation and details on the definitions can be found in the Supplementary Information.

## Data availability

The data supporting the findings of this study are available from the corresponding author upon request.

## Code availability

The code used to perform the simulations of this study is available from the corresponding author upon request.

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

## Acknowledgements

The authors acknowledge funding from the Deutsche Forschungsgemeinschaft (DFG) via the regular project 542873285. Calculations for this research were conducted on the Lichtenberg high-performance computer of the TU Darmstadt (Project 2373).

## Author contributions

G.M. performed the simulations and data analysis. E.M. and S.B. supervised the research.

## Funding

## Competing interests

The authors declare no competing interests.
