## [Transparent Peer Review file · Nature Communications]

Spatiotemporal dynamics of moiré excitons in Van der Waals heterostructures

Corresponding Author: Dr Giuseppe Meneghini

Version 0:

Reviewer comments:

Reviewer #1

(Remarks to the Author)

In this work, Giuseppe Meneghini et al. investigates moiré exciton diffusion using Monte Carlo solutions of the Boltzmann transport equation combined with a continuum model. They identify an intermediate regime (twist angles between 3° and 6°) in which an unexpected enhancement of exciton propagation occurs at lower twist angles and low temperatures. This anomalous behavior is attributed to a relaxation bottleneck caused by the moiré band gap, which hinders excitons from fully dissipating their excess energy. As a result, high-energy excitons accumulate near the flat-band regions. The thermal population of excitons extending beyond these flat regions leads to larger effective group velocities, thereby enhancing exciton diffusion. The results are interesting and sound reasonable. However, the authors should address the following questions before I can recommend publication.

1, The authors employ a continuum model to compute the moiré exciton band structure and exciton–phonon coupling, which inherently involves several approximations. To assess the reliability of these results, the authors should explicitly present the moiré potentials used for different twist angles. Furthermore, they should justify how these potentials are constructed and explain why they are sufficient to capture the essential features of the moiré exciton band structure and the exciton–phonon interactions.

2, In Figure 4, the authors report a decreasing trend of the diffusion coefficient with increasing temperature for the moiré superlattice with a twist angle of 3° , attributing this behavior to a reduction in the scattering time τ . However, the definition of τ is unclear. Is τ referring to the relaxation time of high-energy excitons to the lowest energy states. If so, the computed diffusion coefficient may not fully capture the contribution of phonon-assisted transitions from low-energy to higher-energy states at high temperatures.

3, In the caption of Figure 1, “(a) A twisted TMD heterostructure is optically excited, creating an initial population of hot excitons (red shade) with an energy of 60 meV (highlighted in orange in the excitonic subband structure)” The description of (red shade) and (highlighted in orange in the excitonic subband structure) is not clear in the Figure 1.

4, In Figure 2, what is $R_x(\mu\text{m})$ of the x-axis? Is it excitons width?

5, In Figure 3(b) and (c), the exciton population is shown superimposed on the excitonic subband structure. However, it is unclear whether this population corresponds to an equilibrium distribution or to a specific simulation time step. The authors should clarify this in the caption or main text. Furthermore, the color bar representing the exciton population lacks quantitative information—specifically, the maximum and minimum values are not labeled. Including these values is essential for interpreting the population distribution.

6, In Page 4 line 218, “The smaller the twist angle, the larger the energy gaps and the flatter the bands [Samuels Nano Lett].” The full citation should be added to the reference list.

Reviewer #2

(Remarks to the Author)

The manuscript by Meneghini et al. studies exciton diffusion in vdW heterostructures. They authors develop a microscopic model based on the Boltzmann transport equation and Monte Carlo sampling to calculate the exciton dynamics both in momentum and in real space. They find an enhanced diffusion coefficient due to a non-thermal population of excitons in flat band regions at low temperatures and a certain twist angle.

The manuscript is well written, its structure is very clear, and, as someone with background in exciton physics but not transport theory, the methodology seems sound to me, and the approximations made are meaningful.

While the results and their presentation are clear to me, I am not completely sure I understand the author's interpretation of them (see major points below).

Overall, I think this work is very interesting and could spur experimental and further theoretical studies in this direction, but I'd like to ask the authors to respond to my comments below before I can recommend publication.

Major comments

1. One central argument for the anomalous diffusion coefficient seems to be the non-Boltzmann population of flat regions due to decreased phonon scattering, and the authors write that "the thermal population of extending beyond the flat regions results in larger effective group velocities and explains the increased diffusion coefficient." However, I cannot completely follow this argument for the enhanced diffusion coefficient (also given in lines 193ff and 290ff). They argue that the non-Boltzmann population favors higher group velocities, but looking at Figure 3b, the population is concentrated in the flat regions. Could the authors clarify whether there is a more detailed physical interpretation or whether the increased diffusion coefficient is simply a numerical result?

2. Could the authors comment on the sensitivity of that anomalous diffusion region with respect to the twist angle? They show a selection of 4 twist angles, where only one of them shows that behaviour. Is it also present for a twist angle of, for example, 2 degrees, and where between 3 and 4 degrees would that behaviour change?

3. In line 307f in the Discussion section, the authors write that their study "lays the foundation for a possible design of optoelectronic devices based on moiré excitons.", which sounds a bit vague and generic to me. Could the authors be more specific about what they have in mind here, and how their specific results would have an impact?

4. Due to the permanent out-of-plane electric dipole moment, interlayer excitons can be strongly interacting. As far as I understand, these interactions are not taken into account in the author's theory. Can the authors briefly comment on why this is not included and how they would affect their results?

Minor comments

5. In the caption of Figure 1, I think it should read "blue Gaussian" instead of "red Gaussian" in the second to last sentence.

6. The labels of the x-axes of panels (b) and (c) are given in terms of Momentum (1/nm). However, the ticks only denote the γ and κ points of the mini Brillouin zone, and the moiré period is never mentioned, so I think it is sufficient to just label the axes with "Momentum".

7. In line 219, the citation was not correctly processed.

8. In the Methods section in line 327 (and also in line 335), the index of the exciton operators is omitted but not explained why. This is only done in the SI. I think it would be good to also state this briefly in the Methods to make this clearer.

Version 1:

Reviewer comments:

Reviewer #1

(Remarks to the Author)

I go through the response for our comments and the other reviewer. The authors reply these comments properly. We also thank the authors for careful revisions and responses to our comments. Overall, I suggest the paper can be published after these revisions.

Reviewer #2

(Remarks to the Author)

The authors have addressed all my comments thoroughly in their response and the revised manuscript. Their clarifications and additional analysis have resolved the points where I had questions or found the interpretation unclear. I therefore recommend the manuscript for publication in Nature Communications.

Response to the Reviewers' Comments

We thank the reviewers for carefully reading our manuscript and for their valuable feedback, which helped us to further improve the quality of our manuscript. We have addressed each comment made by the reviewers, providing a detailed point-by-point response below. All changes in the revised manuscript are highlighted in blue.

Reply To Referee 1

Referee:

In this work, Giuseppe Meneghini et al. investigates moiré exciton diffusion using Monte Carlo solutions of the Boltzmann transport equation combined with a continuum model. They identify an intermediate regime (twist angles between 3° and 6°) in which an unexpected enhancement of exciton propagation occurs at lower twist angles and low temperatures. This anomalous behavior is attributed to a relaxation bottleneck caused by the moiré band gap, which hinders excitons from fully dissipating their excess energy. As a result, high-energy excitons accumulate near the flat-band regions. The thermal population of excitons extending beyond these flat regions leads to larger effective group velocities, thereby enhancing exciton diffusion. The results are interesting and sound reasonable. However, the authors should address the following questions before I can recommend publication.

Response:

We are grateful for the overall positive assessment of our work and we appreciate the constructive critique of the referee, which has helped improve the presentation of our results in the revised manuscript.

1. Referee:

The authors employ a continuum model to compute the moiré exciton band structure and exciton-phonon coupling, which inherently involves several approximations. To assess the reliability of these results, the authors should explicitly present the moiré potentials used for different twist angles. Furthermore, they should justify how these potentials are constructed and explain why they are sufficient to capture the essential features of the moiré exciton band structure and the exciton-phonon interactions.

Response:

We thank the referee for this important comment. Although some details have already been provided in the Supplemental Information (SI), we agree that they may not be sufficient to fully reproduce the effective model used in our work. We have now included a more detailed derivation in the SI and added explicit plots of the moiré potentials for different twist angles as suggested by the referee. Furthermore, we have added a clarifying sentence in the Methods section in the main manuscript explaining why the continuum model is valid in the regime considered in our work.

Changes in the manuscript:

We have added the following sentence to the methods section in the main part: "This continuum model accurately captures the low-energy moiré exciton physics in the regime of small twist angles and weak interlayer tunneling, assuming that the moiré potential only slightly affects the intralayer ex-

Figure 1: Real space moiré potential for the different twist angles under study. The potentials are plotted in units of the moiré lattice constant a_M .

citon-phonon interaction, leading mainly to a remodulation of energy and momentum.”. Furthermore, we have added Fig. 1 to the revised SI showing the moiré potential for different twist angles and we have included a new section showing the derivation of the electronic moiré potential: “We know from first-principle calculations that the electron/hole tunneling is negligible around the K valley in the considered TMD heterostructure [1]. In addition, the largest contribution to the electronic states at the K point arises from d -orbitals at the metal atoms, which are protected from the environment by the surrounding chalcogen atoms. To model the twist-angle dependent energy modulation, we therefore assume that the electronic Hamiltonian contains effective electrostatic potentials \hat{W}_l (with $l = 0, 1$ for the two layers) created by the lattices of the two vertically stacked layers yielding

$$H = \sum_{l'l'\mathbf{k}\mathbf{k}'} \langle l\mathbf{k} | \hat{T} + \hat{W}_0 + \hat{W}_1 | l'\mathbf{k}' \rangle a_{l\mathbf{k}}^\dagger a_{l'\mathbf{k}'} = \sum_{l\mathbf{k}} \varepsilon_{l\mathbf{k}} a_{l\mathbf{k}}^\dagger a_{l\mathbf{k}} + \sum_{l\mathbf{k}\mathbf{q}} M_{l\mathbf{k}}(\mathbf{q}) a_{l\mathbf{k}+\mathbf{q}}^\dagger a_{l\mathbf{k}} \quad (1)$$

where $\langle \mathbf{r} | l\mathbf{k} \rangle = \psi_{\mathbf{k}}^l(\mathbf{r}) = \frac{1}{\sqrt{N}} \sum_{\mathbf{R}_i} e^{i(\mathbf{k}+\mathbf{K}_i)\cdot\mathbf{R}_i} \phi_l(\mathbf{r}-\mathbf{R}_i)$ are the monolayer eigenstates at the K point composed of the orbitals ϕ_l at metal-atom sites \mathbf{R}_i . Furthermore, we have introduced the kinetic energy $\hat{T} = \hat{K}_0 + \hat{K}_1$, the monolayer band energies $\varepsilon_{l\mathbf{k}} = \langle l\mathbf{k} | \hat{K}_l + \hat{W}_l | l\mathbf{k} \rangle$, and the moiré potential $M_{l\mathbf{k}}(\mathbf{q}) = \langle l\mathbf{k} | \hat{W}_{1-l} | l\mathbf{k} \rangle$. Expanding the moiré term in atomic contributions $W_l = \sum_{\mathbf{R}_i} w_l(\mathbf{r}-\mathbf{R}_i)$ we can rewrite it as $M_{l\mathbf{k}}(\mathbf{q}) = \sum_{\mathbf{G}_i, \mathbf{G}_{1-l}} m_l(\mathbf{G}_{1-l}) e^{i(\mathbf{G}_i + \mathbf{G}_{1-l})\cdot\mathbf{D}_l/2} \delta_{\mathbf{q}, \mathbf{G}_{1-l} - \mathbf{G}_i}$ with the reciprocal lattice vectors G_l and the atomic interaction energy $m_l(\mathbf{q}) = 1/A_{UC} \int dz \tilde{\rho}_l(-\mathbf{q}, z) \tilde{w}_l(\mathbf{q}, z)$ using $\rho_l(\mathbf{r}) = |\phi_l(\mathbf{r})|^2$ and the in-plane Fourier transform $\tilde{f}(\mathbf{q}, z) = \int d\mathbf{r}_\parallel f(\mathbf{r}) e^{i\mathbf{q}\cdot\mathbf{r}_\parallel}$. Here, A_{UC} is the area of the unit cell and the quantity $\mathbf{D}_l = \mathbf{R}_l^0 - \mathbf{R}_{1-l}^0$ is defined as the spatial displacement of the two lattice origins. Now we exploit that the atomic potentials are smooth functions, so that we can restrict the sum over reciprocal lattice vectors to the first shell. Terms with $\mathbf{G}_l = \mathbf{G}_{1-l} = 0$ lead to stacking- and space-independent band shifts. Furthermore, we can decompose the atomic potential into parts stemming from metal- or chalcogen atoms and exploit the C_3 symmetry of orbitals/atomic potentials in a hexagonal lattice,

which finally yields

$$V_M = \sum_{l\mathbf{k}\mathbf{q}} V_{l\mathbf{q}} a_{l\mathbf{k}+\mathbf{q}}^\dagger a_{l\mathbf{k}} + h.c. \quad (2)$$

$$V_{l\mathbf{q}} = v_l \sum_{n=0}^2 e^{iC_3^n(\mathbf{G}_l+\mathbf{G}_{1-l})\cdot\mathbf{D}_l/2} \delta_{\mathbf{q},C_3^n(\mathbf{G}_{1-l}-\mathbf{G}_l)} \quad (3)$$

with $v_l = \gamma_1^l + \gamma_2^l e^{2\pi\sigma_{1-l}/3}$, where $\sigma_l = 1/-1$ for R-type/H-type stacking configurations. Hence, the potential is fully parametrized by the two numbers γ_1^l and γ_2^l , characterizing the interaction energy of the K-point orbitals with metal- and chalcogen-atoms of the other layer, respectively. These can be directly obtained from first principle calculations by comparing the band energies for different high-symmetry stackings, i.e., by fixing the twist angle to 0° and considering $\mathbf{G}_0 \simeq \mathbf{G}_1$, the potential depends only on the lateral displacements \mathbf{D}_l . As a results, using the energy levels from R_{hh} , R_{Mh} and R_{Xh} allows to unambiguously determine $\gamma_{1/2}^l$ [1]. "

2. Referee:

In Figure 4, the authors report a decreasing trend of the diffusion coefficient with increasing temperature for the moiré superlattice with a twist angle of 3° , attributing this behavior to a reduction in the scattering time τ . However, the definition of τ is unclear. Is τ referring to the relaxation time of high-energy excitons to the lowest energy states. If so, the computed diffusion coefficient may not fully capture the contribution of phonon-assisted transitions from low-energy to higher-energy states at high temperatures.

Response:

We thank the referee for this important comment. The quantity τ refers to the total scattering time, i.e., it is defined as the inverse of the total out-scattering rate, so including all possible relaxation channels. We have added a clearer definition of τ in the revised manuscript and we point out its relation to the total out-scattering rate.

Changes in the manuscript:

Following the referee's suggestion, we have modified the main text by clarifying the connection between the scattering time τ and the total out-scattering rate in line 175: "...and the scattering time $\tau_{\mathbf{k}}^\eta$, obtained as the inverse of the total out-scattering rate from the state η (see supplemental information for more details)".

3. Referee:

In the caption of Figure 1, "(a) A twisted TMD heterostructure is optically excited, creating an initial population of hot excitons (red shade) with an energy of 60 meV (highlighted in orange in the excitonic subband structure)" The description of (red shade) and (highlighted in orange in the excitonic subband structure) is not clear in the Figure 1.

Response:

We thank the referee for raising this point. In Fig. 1, the "red shade" refers to the bubble surrounding the electron-hole pair (the + and - particles). To illustrate hot excitons at the beginning of the relaxation process, we color this bubble red, while blue bubbles represent cold excitons at the end of the relaxation. The "orange" region refers to the plane that indicates the uniform initial distribution of excitons around 60 meV. In the revised manuscript, we have updated the caption of Fig. 1 to clarify these points. Furthermore, we

Figure 2: Schematic of the investigated twisted TMD heterostructure illustrating (a) the exciton band structure and the relaxation of optically excited hot excitons in reciprocal space and (b) the broadening of the spatial exciton distribution due to diffusion in real space.

have also modified the schematic figure to make the distinction between hot and cold excitons clearer, added labels, used stronger colors, and darkened the color of the excitation plane to distinguish it more clearly from the lighter color of the excitonic band structure, cf. Fig. 2 in this response letter.

Changes in the manuscript:

We have rephrased the sentence in the captions of Fig. 1 "A twisted TMD heterostructure is optically excited..." to "A twisted TMD heterostructure is optically excited, creating an initial population of hot excitons (represented by the red contoured and shaded electron-hole pair bubble) with an energy of 60 meV (indicated by the orange trapezoid pointed to by the arrow in the excitonic sub-band structure, representing the uniform initial exciton distribution used in the simulations)." and added "(orange arrows connecting red excitons (hot) to blue excitons (cold))" to connect better to the difference between red shade and blue shade.

4. **Referee:**

In Figure 2, what is $R_x(\mu\text{m})$ of the x-axis? Is it excitons width?

Response:

We refer $R_x(\mu\text{m})$ to the cut of the real space exciton distribution along the x direction. We have included a clearer definition of R_x in the caption of Fig. 2.

Changes in the manuscript:

We have added modified a sentence in the caption of Fig. 2: "Time-dependent real-space cut along the x direction (R_x) of the exciton distribution in an hBN-encapsulated WSe_2 - MoSe_2 heterostructure for a twist angle of 3° and at a temperature of (a) 10 K and (b) 70 K".

5. **Referee:**

In Figure 3(b) and (c), the exciton population is shown superimposed on the

excitonic subband structure. However, it is unclear whether this population corresponds to an equilibrium distribution or to a specific simulation time step. The authors should clarify this in the caption or main text. Furthermore, the color bar representing the exciton population lacks quantitative information- specifically, the maximum and minimum values are not labeled. Including these values is essential for interpreting the population distribution.

Response:

We thank the referee for this important comment. Panel 3(b) and 3(c) show the equilibrium distribution reached after 100 ps in the two different cases of 10 K and 70 K. Since the energy-momentum relaxation occurs in the first 50 ps, the distribution plotted is indeed the equilibrium distribution. We clarify this in the caption of Fig. 3 and include min/max values in the color bar to improve the clarity of the manuscript as suggested by the referee.

Changes in the manuscript:

We have rephrased a sentence in the caption of Fig. 3 "(b)-(c) Exciton occupation at equilibrium (color-coded) overlaid on the exciton subband structure at a temperature of 10 K (b) and 70 K (c)." Furthermore, we have added minimum and maximum values to the color bar to make the population scale quantitative.

6. Referee:

In Page 4 line 218, "The smaller the twist angle, the larger the energy gaps and the flatter the bands [Samuels Nano Lett]." The full citation should be added to the reference list.

Response:

We thank the referee for carefully reading our manuscript and noticing this incomplete citation.

Changes in the manuscript:

We have inserted the correct citation in the revised manuscript.

Reply To Referee 2

Referee:

The manuscript by Meneghini et al. studies exciton diffusion in vdW heterostructures. They authors develop a microscopic model based on the Boltzmann transport equation and Monte Carlo sampling to calculate the exciton dynamics both in momentum and in real space. They find an enhanced diffusion coefficient due to a non-thermal population of excitons in flat band regions at low temperatures and a certain twist angle.

The manuscript is well written, its structure is very clear, and, as someone with background in exciton physics but not transport theory, the methodology seems sound to me, and the approximations made are meaningful.

Overall, I think this work is very interesting and could spur experimental and further theoretical studies in this direction, but I'd like to ask the authors to respond to my comments below before I can recommend publication.

Response:

We thank the referee for the positive assessment of our work and for the time

and effort dedicated to reviewing our manuscript. We found the points raised by the referee very useful and important for improving the overall clarity of our work.

1. Referee:

One central argument for the anomalous diffusion coefficient seems to be the non-Boltzmann population of flat regions due to decreased phonon scattering, and the authors write that "the thermal population of extending beyond the flat regions results in larger effective group velocities and explains the increased diffusion coefficient." However, I cannot completely follow this argument for the enhanced diffusion coefficient (also given in lines 193ff and 290ff). They argue that the non-Boltzmann population favors higher group velocities, but looking at Figure 3b, the population is concentrated in the flat regions. Could the authors clarify whether there is a more detailed physical interpretation or whether the increased diffusion coefficient is simply a numerical result?

Response:

We thank the referee for this important question and the opportunity to clarify this point. Excitons are indeed mostly localized around the flat region along the illustrated high-symmetry path (Γ - K - M in the moiré BZ) in Fig. 3(b). However, the excitonic subband is not flat in all directions in the moiré Brillouin zone, as the band flattens only along specific directions but remains more dispersive along others. This can be seen in the schematic 3D representation of the full exciton band structure shown in Fig. 1. At elevated temperatures, the thermal population extends into the more dispersive directions, allowing excitons to access states with larger group velocities than what is visible along a single cut. Therefore, the increased diffusion coefficient is physically explained by this additional phase space for excitons to scatter into states with higher group velocities, despite the apparent concentration in the flat region along the plotted path. We have clarified this point in the revised manuscript.

Changes in the manuscript:

We have rephrase the paragraph between the lines 190-209 as follows: "At lower temperatures, excitons remain trapped (bottleneck effect) in relatively flat regions of the dispersion landscape (cf. Fig. 3(b) showing the exciton occupation superimposed to the band structure along the path $\gamma \rightarrow \kappa \rightarrow m$), where the mismatch between the interband energy gap and the energies of the dominant optical phonons prevents further relaxation to the ground state via phonon emission [2]. However, the excitonic band structure is not flat in all directions and the thermal population can partially extend into more dispersive regions of the moiré Brillouin zone (cf. the schematic 3D plot of the excitonic bandstructure in Fig. 1), allowing excitons to access states with higher group velocities. This directional extension of the population explains the larger diffusion coefficient compared to the fully thermalized case, despite the apparent flatness seen along the plotted path. This is a counterintuitive result, as one might expect flat bands to trap excitons and hinder their propagation."

2. Referee:

Could the authors comment on the sensitivity of that anomalous diffusion region with respect to the twist angle? They show a selection of 4 twist angles, where only one of them shows that behaviour. Is it also present for a twist angle of, for example, 2 degrees, and where between 3 and 4 degrees would that behavior change?

Response:

We thank the referee for this important comment. At a twist angle of about 2° , the anomalous diffusion behavior is not observed because the lowest moiré exciton bands become nearly completely flat, effectively reducing the group velocity to zero. In this regime of extreme flatness, the Boltzmann transport equation approach cannot adequately capture exciton dynamics and diffusion, since excitons are effectively localized and diffusion is strongly suppressed. For twist angles between 3° and 4° , we observe a gradual onset of the anomalous diffusion regime. This behavior arises from the interplay between the moiré exciton bandwidth of the ground state and the bandgap separating the ground and first excited states. Specifically, when the bandgap remains sufficiently large to suppress acoustic phonon scattering, and the bandwidth of the ground-state band is smaller than the optical phonon energy, a phonon bottleneck emerges. This bottleneck leads to a non-Boltzmann population concentrated near the band minima of the excited state, resulting in an enhanced exciton diffusion.

To clarify this behavior, we have included a new figure showing the twist-angle dependence of the moiré exciton bandwidth and bandgap, together with the optical phonon energy, to highlight the region in which the bottleneck effect is expected to give rise to an anomalous diffusion, cf. Fig. 3 in this response letter.

Changes in the manuscript:

We have included the new paragraph "Anomalous diffusion regime" in the revised SI explaining in more detail the interplay of moiré exciton bandwidth and bandgap including Fig. 3. " As shown in the main text, for a twist angle of 3° and temperatures below 60 K, we observe an anomalous regime of enhanced diffusion caused by a non-Boltzmann distribution of excitons due to a strong phonon bottleneck effect. To understand the possible extent of this regime with respect to twist angle, note that at about 2° the anomalous diffusion behavior does not appear because the lowest moiré exciton bands become nearly completely flat (Fig. 3(b)), effectively reducing the group velocity to zero. In this regime of flatness, the Boltzmann transport equation approach is no longer adequate to describe exciton dynamics and diffusion, since excitons are effectively localized and diffusion is strongly suppressed due to the vanishing group velocity. For twist angles between 3° and 4° , we observe a gradual onset of the anomalous diffusion regime. This behavior arises from the interplay between the moiré exciton ground-state bandwidth and the bandgap separating the ground and first excited states. Specifically, when the bandgap remains sufficiently large to suppress acoustic phonon scattering, and the bandwidth of the ground-state band is smaller than the optical phonon energy, a phonon bottleneck emerges. This bottleneck causes a non-Boltzmann population to accumulate near the band minima of the excited state, resulting in enhanced diffusion coefficients. This is illustrated in Fig. 3(a), which shows the twist angle dependence of the moiré exciton ground-state bandwidth (blue line) and the bandgap between the ground and first excited states (purple line with circles). The orange dashed line indicates the optical phonon energy, while the shaded region highlights the resonance broadening due to dephasing at temperatures around 70 K. The intersection between the lower bound of the orange region and the blue curve marks the upper limit beyond which the bottleneck is suppressed. Based on this, we predict that the anomalous diffusion region extends up to approximately 3.7° , as the bandgap between the ground and first excited states becomes small enough to allow relaxation via acoustic phonons."

Figure 3: (a) Twist angle dependence of the moiré exciton ground state bandwidth (blue line) and bandgap between the ground and first excited state (purple line with circles). The orange dashed line indicates the optical phonon energy, and the shaded region highlights the resonance broadening due to dephasing at temperatures around 70 K. The intersection between the orange region and the blue curve marks the upper bound, where the bottleneck is suppressed. We predict the anomalous diffusion region to extend up to approximately 3.7° , as the bandgap between the ground and first excited states (purple line) becomes considerably small allowing relaxation via acoustic phonons. (b)–(d) Exciton band structures for twist angles of 2° , 4° , and 6° , respectively.

3. Referee:

In line 307f in the Discussion section, the authors write that their study "lays the foundation for a possible design of optoelectronic devices based on moiré excitons." which sounds a bit vague and generic to me. Could the authors be more specific about what they have in mind here, and how their specific results would have an impact?

Response:

We thank the referee for pointing this out. We agree that this sentence was too generic and have clarified it in the revised manuscript. In particular,

our results show that the exciton diffusivity can be tuned by controlling the twist angle and temperature. This tunability opens possibilities for designing moiré-based optoelectronic devices where exciton transport length scales and lifetimes can be engineered for specific applications, such as excitonic interconnects, diffusion-controlled light emission, or energy harvesting.

Changes in the manuscript:

We have replaced the rather generic statement "...lays the foundation for a possible design of optoelectronic devices based on moiré excitons." by "...lays the foundation for the design of optoelectronic devices, in which exciton diffusivity and propagation length can be tuned by varying the twist angle and temperature, enabling a control of the exciton flow that is important for e.g. excitonic circuits, energy funneling, or diffusion-mediated light emission."

4. Referee:

Due to the permanent out-of-plane electric dipole moment, interlayer excitons can be strongly interacting. As far as I understand, these interactions are not taken into account in the author's theory. Can the authors briefly comment on why this is not included and how they would affect their results?

Response:

In this work, we focus on the low-excitation regime, which in a Hubbard-like description corresponds to a pocket filling factor much smaller than one. In this dilute limit, exciton–exciton interactions can be neglected. Additionally, the range of twist angles we consider does not reach the regime of extreme band flattening and full exciton localization, where exciton–exciton interactions can become dominant. At high densities and in the limit of extremely flat bands, the Boltzmann transport approach would no longer be sufficient to accurately describe the exciton dynamics, and additional correlation effects would need to be treated that are beyond our current framework. We plan to address this strongly interacting regime in a future work with an extended theoretical treatment.

Changes in the manuscript:

In the revised manuscript, we have rephrased the sentence in line 85 to "We study the spatiotemporal moiré exciton dynamics in a twisted TMD heterostructure in the low-excitation regime, where the exciton density remains sufficiently low to neglect exciton–exciton interactions."

5. Referee:

In the caption of Figure 1, I think it should read "blue Gaussian" instead of "red Gaussian" in the second to last sentence.

Response:

We thank the referee for pointing this out. We fixed the caption.

Changes in the manuscript:

We have replaced "red Gaussian" with "blue Gaussian" in the caption of Fig. 1.

6. Referee:

The labels of the x-axes of panels (b) and (c) are given in terms of Momentum (1/nm). However, the ticks only denote the γ and κ points of the mini Brillouin zone, and the moiré period is never mentioned, so I think it is sufficient to just label the axes with "Momentum".

Response:

We again thank the referee for this suggestion and we have accordingly modified the figure.

Changes in the manuscript:

We have changed the description of the x-axis in Fig. 3b and 3c by removing "1/nm".

7. Referee:

In line 219, the citation was not correctly processed.

Response:

We thank the referee for noticing the incomplete citation, which we fixed.

Changes in the manuscript:

We inserted the correct citation in the revised manuscript.

8. Referee:

In the Methods section in line 327 (and also in line 335), the index of the exciton operators is omitted but not explained why. This is only done in the SI. I think it would be good to also state this briefly in the Methods to make this clearer.

Response:

We thank the referee for pointing this out. We have now clarified in the methods section that we focus on the excitonic ground state, which justifies omitting the excitonic index.

Changes in the manuscript:

We rephrased line 329 of the main manuscript to "Focusing on the excitonic ground state in the following, we neglect the excitonic index μ ..."